# Changes in Exercise Performance in Patients During a 6-Week Inpatient Psychiatric Rehabilitation Program and Associated Effects on Depressive Symptoms

**DOI:** 10.3390/jfmk9040233

**Published:** 2024-11-13

**Authors:** Philipp Birnbaumer, Claudia Traunmüller, Christian Natmessnig, Birgit Senft, Caroline Jaritz, Sigurd Hochfellner, Andreas Schwerdtfeger, Peter Hofmann

**Affiliations:** 1Institute of Human Movement Science, Sport & Health, University of Graz, 8010 Graz, Austria; philipp.birnbaumer@uni-graz.at (P.B.); peter.hofmann@uni-graz.at (P.H.); 2Institute of Psychology, Faculty of Natural Sciences, University of Graz, 8010 Graz, Austria; andreas.schwerdtfeger@uni-graz.at; 3Private Clinic for Psychiatric Rehabilitation, St. Radegund, Sanlas Holding, 8061 St. Radegund, Austria; c.natmessnig@sanlas.at (C.N.); office@statistix.at (B.S.); c.jaritz@sanlas.at (C.J.); s.hochfellner@sanlas.at (S.H.)

**Keywords:** exercise training, cardiorespiratory fitness, cardiopulmonary exercise test, mental health, psychiatric rehabilitation, psychological distress

## Abstract

**Background/Objectives**: The impact of exercise on affective disorders has been demonstrated in various studies. However, almost no data are available on performance effects. Therefore, this study investigated exercise performance related to the severity of depression symptoms in a pre–post within-subjects design in a 6-week standard inpatient psychiatric rehabilitation program. **Methods**: A total of 53 individuals (20 female; mean age, 40.98 ± 11.33) with a primary diagnosis of depression performed a cardiopulmonary exercise test (CPX) to obtain maximal oxygen uptake (VO_2max_), maximal power output (P_max_), and the first and second ventilatory threshold (VT_1_, VT_2_) at the start and the end of the rehabilitation. Degree of depression was assessed by Becks Depression Inventory (BDI) and the Brief Symptom Inventory test (BSI). Overall activity was measured by accelerometer step-counts. **Results**: Mean total step-count per day during rehabilitation was high (12,586 ± 2819 steps/day). Patients’ BDI and BSI at entry were 21.6 ± 8.83 and 65.1 ± 6.8, respectively, and decreased significantly (*p* < 0.001) following rehabilitation, to 10.1 ± 9.5 and 54.5 ± 11.3, respectively. P_max_ and VO_2max_ increased significantly (*p* < 0.001) from entry values (182.6 ± 58.7 W, 29.74 ± 5.92 mL·kg^−1^·min^−1^) following rehabilitation: by 11.91 ± 12.09 W and 1.35 ± 2.78 mL·kg^−1^·min^−1^, respectively. VT_1_ and VT_2_ showed a similar behavior. An increase in physical performance could predict improvement in BDI (R^2^ = 0.104, F(1,48) = 5.582, *p* = 0.022) but not in BSI. **Conclusions**: The program was associated with improved mental health status in all patients and increased performance in the majority of patients, although increases were small. Since improvements in exercise performance may be positively related to depression symptoms and comorbidities, it is recommended to individualize and tailor exercise programs, which could yield larger effects.

## 1. Introduction

Based on Kessler et al. [1] and according to the “Health at a Glance Europe 2018” report, approximately 84 million people in the EU are affected by mental health problems, with both depression and anxiety disorders being the first and sixth leading causes of disability, respectively [2]. The “Health at a Glance Europe report 2022” showed that almost one in two young Europeans reported a need for psychosocial support [3]. In addition to the personal suffering of those affected, mental health problems have a considerable financial impact on society. The total costs of mental illnesses are more than 4% of the GDP of the 27 EU countries and the United Kingdom, or over EUR 600 billion [3]. Apart from this, the mortality rate for people with mental illness is two to three times higher as compared to the general population [4,5], thus shortening life expectancy up to 15 years [6,7,8]. This is partly due to people with affective disorders showing an increased risk of developing a metabolic syndrome [9] and type 2 diabetes mellitus [10]. To counteract this development, an active lifestyle and regular physical exercise seem to be effective methods [11,12]. Reported benefits of physical activity for people with mental illness are symptom reduction, improved cardiovascular risk profile, and improved physical capacity [13]. Furthermore, an active lifestyle has been linked to mental health and well-being [14]. Stanton et al. [15] highlighted the benefits of physical activity for individuals with mental health problems alongside pharmacological and psychological interventions. However, it has been shown that psychotropic medication (antipsychotics, antidepressants, and mood stabilizers) was associated with an increased risk for several physical diseases [16], thus highlighting the importance of physical activity and exercise as an effective complement in the treatment of mental diseases.

Several meta-analyses indicate antidepressant effects of physical activity in people with depression [17,18,19,20]. For this reason, the European Psychiatric Association identified physical activity as an important primary or adjunct strategy in the treatment of depression [21]. Heissel et al. [22] confirmed the role of physical exercise for depressive disorder and recommend that exercise should be offered as an evidence-based treatment option, supervised, and offered as a group exercise at moderate aerobic intensity. However, it was also mentioned, critically, that more high-quality trials are needed to determine valid parameters for physical performance, as well as to evaluate programs to motivate patients to maintain their exercise after discharge from a hospital setting [23].

The level of cardiorespiratory fitness (CRF) is an important measure of physical status and a strong predictor of all-cause mortality [24]. CRF level is also associated with the risk of common mental health disorders [25]. With regard to the call for evidence-based exercise treatment, only few data are available on CRF and performance measures related to health-related variables in patients diagnosed with affective disorders. This is somehow surprising, as the evaluation of the current physical performance level of individuals builds up the basis for target-orientated and efficient exercise treatment. Herbsleb et al. [26] showed that CRF was expected to be significantly lower in patients with major depression (MDD) compared to healthy controls, indicated by lower peak oxygen uptake and peak power output, similar to Gerber et al. [27], who showed a lower CRF in patients with first or recurrent depression and bipolar disorder type II (currently depressed, based on the International Classification of Diseases, 10th Revision [28]) compared to healthy controls. Most recently, Kreppke et al. [29] also showed that among patients with major depressive disorder, those with particularly high self-assessed severity scores showed lower CRF and less favorable perceived fitness compared to healthy controls. It is difficult to state whether a lower CRF is a consequence or the cause of depression, as both factors influence each other in a vicious circle. However, patients with good CRF were shown to have more favorable scores across cardiorespiratory risk markers (e.g., high body fat, hypertension) than counterparts with poor CRF.

Aiming for higher CRF in people with mental illness, exercise interventions were shown to have a moderate effect on increasing CRF to clinically relevant levels among people experiencing depression [30,31]. Studies showed that regular exercise increases CRF [31,32] and can even lead to a significantly stronger reduction in depressive symptoms compared to usual care without regular exercise [32]. Additionally, physical fitness was shown to be a stronger predictor for the onset of depressive symptoms compared to being overweight or obese [33]. CRF, therefore, appears to be a central resource in the prevention and treatment of depressive disorders and other mental illnesses. However, to the authors’ knowledge, there is yet no clear connection between changes in CRF and depression scores [26]. In addition, there is a well-known relationship between CRF and several chronic diseases, such as cardiovascular or metabolic diseases, which are often combined with depressive disorders [9,13].

Although a growing body of literature supports the notion that exercise and physical activity interventions have beneficial effects across several physical and mental-health outcomes [30,33,34], almost no data are available on exercise performance, its changes, and its association with changes in mental health during a standardized psychiatric rehabilitation program of patients with mental disorders. However, this is of great interest, as the necessity of a stay in a psychiatric rehabilitation increased locally by +21.6% compared to 2019 and by +29.1% compared to 2020 [35], which is also shown in recent European statistics [36]. These numbers not only represent the increasing need for psychiatric rehabilitation but also reflect the rise in mental health problems in the population. This makes the development of targeted and sustainable measures increasingly important. Therefore, the aim of the study was to investigate exercise performance at the beginning of a rehabilitation program, as well as changes in performance during a 6-week standardized rehabilitation program in a specialized rehabilitation clinic and to relate the changes to clinical outcomes and the severity of depression symptoms. We hypothesize that people with a lower depression score have better physical performance than people with higher depression symptoms, as well as that the standard rehabilitation program will not lead to performance increases. Furthermore, we were exploratively interested whether patients with a psychiatric diagnosis are willing to safely push themselves physically to their personal limit during exercise testing.

## 2. Materials and Methods

### 2.1. Procedure

This study was part of a larger project investigating physiological performance data, hemodynamic reactivity data, autonomic physiological regulation parameters, and the total amount of physical activity, which were collected from patients undergoing a 6-week inpatient psychiatric rehabilitation program. The main goal of this project was to see the “real life effects” of rehabilitation, especially focusing on physical performance, as well as the feasibility of exercise testing in psychiatric patients in order to be able to prescribe the exercise parts of rehabilitation in a follow-up. The focus of the present paper was on exercise performance related to the severity of depression symptoms, based on a pre–post within-subjects design. Participants were enrolled in a local rehabilitation center specialized in psychiatric illness and where, on average, 25 patients arrived weekly between May 2022 and February 2023. In rehabilitation, patients are admitted as inpatients for a stay of 6 weeks. During this time, patients are not allowed to leave the rehabilitation center without the consent of a physician. In order to be able to undergo psychiatric rehabilitation, an application must be submitted to the insurance institution by the relevant specialist or general practitioner. Only after a positive decision can a person register for the next possible appointment. The current waiting time for a free place is more than 12 weeks, on average. In principle, patients can indicate a preference as to which facility they would like to go to. However, the next available place will be allocated according to urgency. The team in each facility is multidisciplinary and consists of psychiatrists, psychotherapists, clinical psychologists, ergo-therapists and sport scientists accredited for training therapy. Further information regarding the therapy plan is presented in Table 1.

### 2.2. Participants

Recruitment was carried out by the clinic’s physicians, who allocated 2–3 patients fulfilling the study inclusion criteria on a weekly basis. Only patients staying for the complete 6 weeks program were included in the study. Exclusion criteria were cardiovascular diseases (e.g., prior myocardial infarction or stroke) including medication, for example, to treat blood pressure in order to minimize risks with respect to a maximal incremental exercise test [27]. Also, patients who could not be expected to perform a test on the cycle ergometer due to orthopedic problems were excluded from the study. In a second step, we included patients who received an explanation of the study with detailed information by an accredited exercise scientist (training therapist), the medical director, and the study management the day after. Confidentiality, anonymity, and the opportunity to withdraw from the study at any time without any negative repercussion were assured. Participants who fulfilled all inclusion criteria read and signed an informed consent form. The study was approved by the local ethics committee (GZ. 39/42/63 ex 2019/20).

In total, 63 patients participated in the study. During the study, 7 participants dropped out due to illness not related to the study. The analysis of this study was restricted to patients whose medication remained unchanged during the stay and to patients with psychiatric diagnoses F3 (affective disorders) and F4 (neurotic, stress, and somatoform disorders) according to the International Classification of Diseases, 10th Revision [28]. Three participants with other diagnoses were excluded. Finally, 53 (20 female) patients, of whom 66% had an F3 diagnosis (ICD F31, bipolar affective disorder; F32, depressive episode; F33, recurrent depressive disorder) and 34% had an F4 diagnosis (ICD F40, phobic anxiety disorders; F41, other anxiety disorders; F43, reaction to severe stress and adjustment disorders; F45, somatoform disorder), and who completed the 6-week rehabilitation program and performed an incremental exercise test at the beginning and end of their stay were analyzed. The demographic data, including medication, are shown in Table 2.

### 2.3. Therapy Program 

The scope of the therapy for each patient includes an average of at least 4 h of therapy per day prescribed from Monday to Friday. The number of overall therapy hours, which is obligatory, is between 124.0 h and 126.7 h depending on the arrival of the patient. Patients in the sample received 124 h of therapy sessions. Daily minimum therapy duration was 120 min from Monday to Friday and 60 min on Saturday. Therapy sessions—including physical activity such as strength and endurance-type exercise and training, sports games, and physical coordination tasks—cumulated to 150 min per week (Session 2, Table 1). In general, the mixture of therapy sessions for each patient was built up by different therapeutic content groups in a specific frequency, as shown in Table 1. Exercise sessions were not monitored for intensity, and the overall intensity was rated low to moderate in line with the lower end of the actual physical activity recommendations [37]. Activity measures were determined from wrist-worn accelerometer with the ActiGraph wGT3X-BT (ActiGraph, Pensacola, FL, USA) using the total number of steps per day during the rehabilitation stay. Participants were asked to wear the accelerometer on their non-dominant wrist on every day of the week, including the weekends, from the time they got up in the morning (usually 6 am) until they went to bed at around 10 pm (mandatory room rest). The total number of steps per day includes weekdays, as well as weekends, measured during the rehabilitation stay, with a wear time of at least 720 min according to the ActiLive v6.8.2 Software (ActiGraph, Pensacola, FL, USA) wear time analysis tool.

### 2.4. Cardiopulmonary Exercise Test (CPX)

To obtain objective measures of cardiorespiratory fitness (CRF), patients included in the study performed a standardized cardiopulmonary exercise test (CPX) at the beginning and at the end of the 6-week rehabilitation program. All participants performed a maximal incremental exercise test until volitional exhaustion on a standard cycle ergometer (eBike II Basic & BP, GE Healthcare, Chicago IL, USA) with continuous measures of expired air with a breath-by-breath system (Vyntus CPX, Vyaire Medical, Mettawa, IL, USA) and ECG measures (CardioSoft ECG, GE Healthcare, Chicago IL, USA). The exercise protocol started with a three-minute resting period followed by a three-minute warm-up phase at 20 W. Subsequently, the load was increased by 10 W, 15 W, or 20 W, depending on the investigator’s opinion of the participants’ fitness level, in order to reach maximal exercise at volitional exhaustion in approximately 15 min in all subjects. The protocol followed the standard guidelines of the local cardiology society [38]. Pre- and post-test were performed applying the same protocol. The highest average oxygen consumption measured in the CPX over a period of 30 s was used to determine the maximal oxygen uptake (VO_2max_). The first threshold of ventilation (VT_1_) was determined as the first increase in ventilation (VE) accompanied by a minimum of the oxygen equivalent (VE/VO_2_) without an increase in the carbon dioxide (CO_2_) equivalent (VE/VCO_2_). The second ventilatory threshold (VT_2_) was determined as the second distinct increase in VE accompanied by an increase in both VE/VO_2_ and VE/VCO_2_ [39]. All thresholds were detected by computer-aided multilinear regression analyses applying the Vienna CPX-Tool v1.1.2 (University of Vienna, Vienna, Austria). To adjust for age and to compare performance to average values in healthy individuals, maximal power output (P_max_) was also expressed as a percentage of age-predicted performance, which was calculated according to guideline references [38] for male and female individuals. 

### 2.5. Clinical Measures

The severity of depression was assessed at the beginning and at the end of the 6-week rehabilitation program applying the “Becks Depression Inventory—BDI-II” [40], which is one of the most widely used self-assessment tools for the severity of depressive symptoms. The items are rated on a scale from 0 to 3. An example question is: “I have as much energy as usual” (0), “I have less energy than usual” (1), “I have so little energy that I can hardly get anything done” (2), or “I have no energy left to do anything at all” (3). Reliability for this scale (Cronbach’s Alpha = 0.85) was very good. Additionally, the short form of the “Brief Symptom Inventory—BSI-18” [41], measuring psychological distress through symptoms of depression, anxiety, and somatization by 18 questions, was applied. This method is widely used in the psychiatric field. The items are rated on a scale from 0 (not at all) to 4 (very strongly) for the last 7 days. Reliability for this scale (Cronbach’s Alpha = 0.81) was very good. 

### 2.6. Statistics

To support the assumption of normally distributed data, parametric analysis and Q–Q plots were used, and data were tested by the Shapiro–Wilk test. Data are shown as mean ± standard deviation (SD). Differences between pre- and post-tests of the anthropometric, cardiorespiratory fitness, and clinical measures data were tested by paired samples *t*-tests. Differences between male and female participants for changes between pre- and post-tests were analyzed using an independent samples *t*-test. Effect size was calculated as Cohen’s d for significant differences between pre- and post-tests. Linear regressions were calculated to evaluate the relationship between P_max_ and BDI, as well as BSI, at the start of the rehabilitation stay, to evaluate the changes in BDI and BSI from pre to post, as well as to evaluate the relationship between changes in performance markers (P_max_, P_VT1_, P_VT2_) of the CPX test and changes in the severity of depression from pre- to post-tests. All statistical analyses were performed in SPSS v29.0.0.0 (IBM Corporation, Armonk, NY, USA). Graphical representations were created with Prism 8.0 (GraphPad Software, Boston, MA, USA). Statistical significance was considered as *p* < 0.05 (two-tailed).

## 3. Results

### 3.1. Cardiorespiratory Fitness 

Descriptive statistics for males and females, as well as their cardiorespiratory fitness parameters from the pre- and post-CPX tests, are shown in Table 3. At the start of the rehabilitation program, participants archived a mean VO_2max_ of 29.74 ± 5.92 mL·kg^−1^·min^−1^ (min/max: 18.00/48.77 mL·kg^−1^·min^−1^) and a P_max_ of 182.64 ± 58.73 W (min/max: 77/335 W), which was 103 ± 16% (min/max: 62/153%) of age- and gender-predicted performance (Figure 1). The number of lower-than-normal-performance scores (lower than 100%) was N = 26 (49%; 62% male, 38% female), and the number of higher-than-normal performance scores (higher than 120%) was N = 9 (17%; 67% male, 33 female). Maximal exhaustion during the CPX test was reached in almost all individuals, indicated by a respiratory exchange ratio (RER) of 1.16 ± 0.08 and reaching 101± 7% of the age-predicted maximum heart rate (Figure 1). Six individuals presented an RER lower than 1.1 at P_max_ at the start of the study and at the end of the study, but RER and BDI were not significantly related pre-test (r = 0.234, *p* > 0.092) and post-test (r = 0.085, *p* = 0.633), respectively. However, pre- and post-test RER was highly reproducible (r = 0.712, *p* < 0.001). At the end of the rehabilitation program, mean VO_2max_, and P_max_ were significantly improved by 1.35 ± 2.78 mL·kg^−1^·min^−1^ and 11.91 ± 12.09 W (7 ± 6%), respectively. There was no change in P_max_ in 16 participants (30%), a decrease in 2 participants (4%), and an increase in 35 (66%) participants. Also, mean power at VT_1_ and VT_2_ significantly increased by 4.30 ± 7.23 and 7.95 ± 9.40 W, respectively, in post- compared to pre-tests. Strong effects were shown for changes in P_max_ and power at VT_2_, and a medium effect was shown for power at VT_1_, as well as a small effect for VO_2max_. Significant changes for BDI, BSI, P_max_, and P_VT2_ were found independent of sex in male and female participants but not for VO_2max_ (Table 3). Pearson correlation revealed no relationship between age-predicted performance at the start of the rehabilitation program and a relative performance increase during the rehabilitation program (r = 0.068, *p* = 0.627). However, classification by age-predicted performance quartiles showed a higher increase in the low and high quartile of 8 ± 7% and 9 ± 6% compared to the second and third quartile, both with 5 ± 6%. Despite an increase in power, HR_max_, as well as mean HR and VO_2_, remained unchanged at VT_1_ and VT_2_ (Table 3). Furthermore, body mass index was normal in female participants and close to normal in male participants, while no changes were observed due to the rehabilitation program. Activity measures revealed a mean number of total steps per day during the rehabilitation stay of 12,586 ± 2819 steps. There was no statistical difference between male (12,936 ± 2817 steps) and female (12,032 ± 2806 steps) participants.

Mean changes between pre- to post-tests in VO_2max_, HR_max_, P_VT1_, and P_VT2_, as well as HR and VO_2_ at VT_1_ and VT_2_, respectively, were not statistically different between male and female participants. P_max_ and P_VT1_ were significantly increased in the post-test in male and female participants, as well as P_VT2_, but only significantly in male participants. However, P_max_ was significantly less increased in female participants as compared to male participants (Table 3). Figure 1 shows HR, VO_2_, and VE at the end of a 20 W warm-up period and VT_1_ and VT_2_ at the maximum intensity of the CPX pre- and post-tests for male and female participants. Figures for HR show a clear right shift in the mean curves at maximal and submaximal values, indicating some training adaptation effects.

### 3.2. Clinical Measures

BDI at the beginning of the stay was 21.1 ± 8.6 (min/max: 3/39) and significantly (*p* < 0.001) decreased to 10.8 ± 9.0 (min/max: 0/36) at the end of the rehabilitation program. BSI was 65.1 ± 6.8 (min/max: 49/80) at the start and significantly (*p* < 0.001) decreased to 54.53 ± 11.26 (min/max: 30/72) at the end of the rehabilitation program. Changes in BDI, as well as in BSI, were not statistically different between male and female participants. Pearson correlations showed significant relationships between BDI and BSI at pre- (r = 0.709, *p* < 0.001) and post-test time points (r = 0.740, *p* < 0.001).

### 3.3. Relationship of Clinical Measures and Cardiorespiratory Fitness

P_max_ at the start of the rehabilitation was not significantly related to the BDI (R^2^ = 0.028, F(1,51) = 1.457, *p* = 0.233) or BSI (R^2^ = 0.034, F(1,51) = 1.802, *p* = 0.185) scores, as well as to changes in BDI (R^2^ = 0.003, F(1,48) = 0.167, *p* = 0.685) or BSI (R^2^ = 0.008, F(1,49) = 0.404, *p* = 0.528) from pre- to post-test. Additionally, either the BDI or the BSI score at the start of the rehabilitation could predict changes in P_max_ from pre- to post-test (BDI: R^2^ = 0.06, F(1,51) = 3.21, *p* = 0.078; BSI: R^2^ = 0.029, F(1,51) = 1.512, *p* = 0.224). To relate performance changes to the severity of depression symptoms following the stay, linear regression was calculated for changes in P_max_ and BDI, as well as P_max_ and BSI. A significant regression was found for BDI (R^2^ = 0.104, F(1,48) = 5.582, *p* = 0.022) but not for BSI (R^2^ = 0.004, F(1,49) = 0.188, *p* = 0.667). Changes in P_max_ (B = −0.227, 95% CI −0.420–−0.034) were significantly associated with changes in BDI and explained approximately 10% of the variance in the changes in BDI. The constant term for changes in the BDI score was −8.80 (with minus values indicating improvements), which means an improvement in the BDI score by 0.23 points for every single watt of improvement in P*_max_* (Figure 2A,B).

Additionally, pre-to-post rehabilitation changes in power output at VT_1_ (R^2^ = 0.147, F(1,48) = 8.254, *p* = 0.006) and VT_2_ (R^2^ = 0.126, F(1,48) = 6.950, *p* = 0.011) were shown to be significantly related to changes in BDI score (PVT_1_: (B = −0.450, 95% CI −0.765–−0.135), PVT_2_: (B = −0.326, 95%CI −0.574–−0.077)).

## 4. Discussion

In the present study, we explored cardiorespiratory fitness (CRF) in patients with a primary diagnosis of depression at the beginning and at the end of their 6-week stay in an Austrian rehabilitation center specialized in psychiatric illness and its association with the severity of depression symptoms. Overall, we could show that CRF significantly increased, and depression score and physiological distress significantly decreased, after the end of the stay. Additionally, increased performance was significantly related to improvements in the severity of depression symptoms. Against our two hypotheses, the severity of depression was not related to the performance level at the start of the rehabilitation and performance increased significantly during the rehabilitation program.

Notably, a maximal incremental exercise test to determine CRF was feasible in all participants included in the study, independent of their physical status and the severity of their mental disorder, with 89% of the participants reaching maximal volitional exhaustion according to standard criteria. A similar finding was reported by Boettger et al. [42], indicating a mean RER of 1.11 following a standardized maximal incremental exercise test on a cycle ergometer in 15 patients with major depressive disorder. This is not limited to depression only but has also been demonstrated in patients with other mental disorders like bipolar disorder [43] or schizophrenia [44].

Patients’ performance levels in our study at the start of the rehabilitation were quite heterogenous though, with almost half of the patients performing below and almost half within the age- and gender-predicted norm levels of performance for incremental cycle ergometer exercise [38]. Only nine patients performed above predicted normative values. Mean VO_2max_ was comparable to a sample of male and female patients with major depressive disorder [42]. The depression score of most patients in our study was in a mild to moderate range; five individuals reached values below a clinically relevant score (BDI ≤ 12) and eleven fulfilled the criteria of a major depression (BDI ≥ 29). Our hypothesis that people with a lower level of depression show higher CRF at the beginning of the rehabilitation stay could not be confirmed. This was somehow surprising and unexpected, as some reviews found the severity of depression to be inversely correlated with CRF [25,45]. These authors also highlighted CRF as a valid and important parameter for identifying and preventing mental health disorders. A study by Thirlaway et al. [46] confirmed such an inverse association between CRF and depressive symptoms but only for those who were physically inactive, not for those who were at least moderately active. The authors suggested that not only the level of CRF determines such an inverse relationship with depression but also the current individual activity level. We may therefore suggest that acute effects of physical activity and exercise are the main influence on depressive symptoms, explaining the finding of no relation between CRF and the severity of depression in our results, although we did not obtain home-based activity measures before the rehabilitation stay. 

It is important to note that the performance levels of patients (and, of course, in general) were rather different at the start of the rehabilitation, while the exercise program during rehabilitation was the same. To achieve training effects, the intensity of exercise should generally be above a certain individual level. However, the current exercise program lacks individuality, which may lead to patients being either over- or underchallenged in such a standardized program. It is important to mention that, compared to underchallenge, an overchallenged situation presents some more risks and could have negative consequences especially in individuals suffering from a chronic disease [47]. The relevance of our results is additionally underpinned by scientific results that go beyond the basic correlation between physical activity and mental health [48], focusing more on the impact of the intensity of physical activity and exercise on mental health. A recent review on the effects of exercise on depression [49] showed a clear dose–response relationship for exercise intensity across exercise modalities. Some modalities (yoga, walking, or strength training, among others) showed higher effects in men or women or depending on age, but benefits tended to be proportional to the intensity only. This is in line with other studies, where the highest effects have been observed at moderate and high intensity, but not at low intensity, of physical activity [50,51,52]. In order to utilize the optimum effect of exercise and training in a rehabilitation setting, our data support the need to determine the physical performance of each patient at the beginning of their program to be able to adjust the intensity of the training individually with respect to a personalized medicine approach [53]. 

Although there was no individualized exercise program in our study and the time span of 6 weeks was relatively short, with low to moderate intensity on average, performance in maximal incremental cycle ergometer exercise was significantly increased after the stay. In greater detail, performance increased in the majority (66%) of the patients, stayed the same in 30%, and decreased in 4%. Also, performance at sub-maximal values increased and the mean heart rate curve (Figure 2) showed a clear right shift from pre- to post-test, confirming a performance increase due to training adaptions, thus excluding motivational factors or day-to-day variability. However, increases in VO_2max_ were lower than the suggested minimal clinically important difference (MCID) of at least 3.5 mL·min^−1^·kg^−1^ [54]. The performance increase was similar at VT_1_ and VT_2_ and increased across the entire performance spectrum. This was to be expected, since, as already mentioned, there were no specific individual training sessions focusing on specific adaptations, like an increase in the aerobic or anaerobic performance. This could be of interest, as Chu et al. [55] showed that higher-intensity (65–75% MaxVO_2reserve_) compared to lower-intensity (40–55% MaxVO_2reserve_) exercise training once a week for 30–40 min for 10 weeks led to a comparable decrease in the BDI score, and the higher-intensity aerobic exercise program appeared to be more effective in reducing symptoms of depression. The increase in performance in our study was likely caused by the high total amount of physical activity per day as indicated by the high number of total steps per day. Completing the daily therapy plan alone involves a high level of activity, as the different therapies take place in different rooms and on different floors. Patients reported that they moved significantly more every day than in their daily routine. In addition, they reported to be much more active in their free time, as the rehabilitation center is located in the countryside and offers patients the opportunity to go for a walk outside. 

Additionally, it was found that the pre-to-post-rehabilitation changes in maximal performance were significant but independent of the individual performance level at the start of rehabilitation. However, the increase in performance was greater in the 25% lowest-performing, as well as the highest-performing, patients. This can be explained by the fact that less-fit individuals have a higher potential to increase performance compared to individuals with a higher fitness level or with a larger number of exercise or training hours completed (ceiling effects). Hence, although the fittest individuals should increase less, they increased more as compared to the mean. This could be explained by the fact that the physically fittest 25% of the patients were still not well-trained individuals and still had a high potential for development; in addition, this group might have a higher motivation and therefore work harder during the sessions as compared to the others. It is important to note that 30% of patients did not improve their physical performance. An individualized training program might have led to a lower percentage of non-responders to the standardized exercise program, although intensity was not monitored during the exercise sessions. 

VO_2max_ also significantly increased, by around 1.35 mL·min^−1^·kg^−1^ (3.29%) at constant body weight, which is more than double the growth shown after a supervised aerobic exercise intervention of 90 min of training per week [31]. Looking at the VO_2peak_ trainability of structured interval and moderate continuous training, a multi-center comparison from Wiliams et al. [54] presented an increase of 2.4 mL·min^−1^·kg^−1^ for continuous training and of 3.3 mL·min^−1^·kg^−1^ for interval training when results were, among others, adjusted for sex, age, and number of training sessions. Both individualization and a more structured training concept offer potential for an even greater increase in VO_2max_ within the stay. Since the VO_2max_ increase within the rehabilitation was markedly lower than the MCID, which is associated with a 10–25% decreased risk of all-cause mortality [56], adaptation of the exercise concept could be of interest. 

Regarding changes in mental health, our data show overall decreased depression symptoms, from moderate depression at the beginning to minimal depression symptoms at the end of the rehabilitation stay. A comparable finding could be shown for BSI, which also indicated a significant reduction in patients’ subjective impairments due to physical and psychological symptoms. It has already been shown that exercise significantly increased cardiorespiratory fitness in people with depression [30] and positively influenced the severity [32,55]. Stubbs et al. [30] also showed a statistical trend for higher baseline depressive symptoms to predict smaller increases in CRF following 12-week-long, planned, structured, and repetitive exercise interventions. This is contrary to our findings, where the increase in performance was not related to the severity of depression symptoms at the start of the rehabilitation program. However, an increase in performance was shown to be not necessarily accompanied by a reduction in the severity of depression [31]. It may be suggested that the effects of exercise are not only due to the activity itself but are also influenced by other factors that go hand in hand with it. Noetel et al. [49] already hypothesized that a combination of social interaction, mindfulness or experiential acceptance, increased self-efficacy, immersion in green spaces, neurobiological mechanisms, and acute positive affects due to exercise combine to generate outcomes. Many types of exercise cover some of these factors but no single one covers all mechanisms. 

However, in our study, linear regression analysis showed that changes in P_max_ could predict significant changes in BDI but not for BSI scores. Looking at the relative changes, we found an improvement of approximately 53% in the BDI but only of 16% in the BSI score, causing less statistical power possibly explaining the effects. For BDI, each watt of improvement in P_max_ was related to an improvement in the BDI score. However, the linear regression coefficient was small, and the model only explained 10% of the variance. Interestingly, performance changes at VT_1_ and VT_2_ were more strongly related to changes in BDI score compared to P_max_. Therefore, these values seemed to be a better predictor of mental health, with the advantage that no maximal cardiorespiratory test must be carried out. A study by Blumenthal et al. [32] already showed significantly stronger improvements in BDI score when patients engaged regularly in physical exercise (90 min/week) for three months in addition to their usual treatment relative to not exercising. Furthermore, patients with a BDI score ≥ 14 were shown to benefit even more from additional regular exercise as compared to individuals below this score. However, this study did not investigate the relation between changes in physical performance and changes in BDI. Regarding this relation, there are also results complementary to ours showing a substantial increase in performance (15% in P_max_/kg, 19% in VO_2max_/kg) and a decrease in the Hamilton-17 depression scale by 2–3 points, independent of the increase in performance [31]. At present, no clear causality can be established between physical performance changes and the severity of depression symptoms. In general, performance improves as a result of regular exercise. As already mentioned, it is unclear to what extent performance-related physiological adaptations or social and environmental effects of exercise influence changes in mental health. Also, an improved mental health status following six weeks of rehabilitation is a sum effect of all therapies. Each specific effect of each specific therapy cannot be determined in such a real-world approach. It should be emphasized, however, that performance is a resource which can be developed to a considerable degree by well-known methods. This offers great potential to improve the mental health status of individuals with depressive symptoms and even greater potential for reducing comorbidities [24,36].

In addition to general effects, several studies also showed biological effects of exercise training related to the disease. Recently, Gujral et al. [57] showed that acutely depressed adults presented regional gray-matter abnormalities, as well as abnormalities in other brain regions. Exercise was shown to positively influence these structures, e.g., hippocampal volume, and the evidence was described as promising. Although these authors reported inconsistencies between types and duration of exercise, precluding a clear understanding of the underlying effects, they concluded that moderate-intensity aerobic exercise, as a non-pharmacological strategy, improves hippocampal volume, supporting the assumption of intensity as an important variable. Additionally, evidence was reported for a positive relationship between CRF, physical activity, and prefrontal cortex volume. Similar effects were prescribed for striatum and white matter integrity by these authors [57]. In addition to the prescribed effects of exercise on brain structures, it was recently shown by Ross et al. [58] that acute bouts of exercise are able to modulate circulating levels of serotonin and norepinephrine, brain-derived neurotrophic factor, and a variety of immuno-inflammatory mechanisms in clinical cohorts with depression. However, these authors argue that chronic exercise training has not been demonstrated to consistently modulate such mechanisms, and evidence linking these putative mechanisms and reductions in depression is lacking. They also highlighted that the complexity of the biological underpinnings of depression coupled with the complicated molecular cascade induced by exercise are significant barriers to understanding the effects of exercise on depression. Despite these limitations, clinical evidence uniformly argues for the application of exercise to treat depression [58]. 

Nevertheless, more studies focusing on exercise effects are of potential interest. In particular, randomized controlled trials investigating these effects in a rehabilitation setting, as well as in patient care, appear mandatory.

### Limitations

Some limitations of the study need to be mentioned. Our results are limited to individuals in a rehabilitation setting and cannot be transferred to those with an acute status of affective disorders; those with neurotic, stress, and somatoform disorders; or individuals with overly strong depressive symptoms. Furthermore, patients were pre-selected before they were included in the study, at the start of rehabilitation, in order to minimize cardiac risk. Therefore, individuals with cardiovascular or other diseases are not represented in the actual analysis. However, although participants were pre-selected, there was high heterogeneity in exercise performance, as well as in the severity of depression symptoms. Inclusion of patients with comorbidities will be important for future studies and may allow more general conclusions. We also did not analyze the effects of medication in combination with the rehabilitation program on the improvement in depression. However, as medication did not change, we suggest that the improvements were due to the program (exercise and other therapies) but cannot exclude interactions between those factors. Regarding the evaluation of mental health, the self-assessment tools used to evaluate the severity of depression (BDI and BSI) are partly a limit compared to a more objective assessment by a physician. Furthermore, there was no control of the intensity of the standardized exercise program. Monitoring the intensity of the sessions would have allowed us to assign performance effects, as well as changes in mental health status, to specific exercise and training stimuli. Future studies should focus on this in order to determine more precisely the cause of the physical and mental adaptations during rehabilitation. It should also be mentioned that in the present study we did not record the status of physical activity prior to the rehabilitation stay, which would have provided additional information about the relationship between CRF and the severity of depressive symptoms. Furthermore, a control group is missing, meaning patients who are registered for a rehabilitation stay but are on a waiting list. This was not possible due to the current data protection regulations.

## 5. Conclusions

This study showed that even in patients with psychiatric diagnoses of affective disorders and neurotic, stress, and somatoform disorders, it seems feasible to safely perform a reproducible maximal incremental exercise test. Overall, the program was associated with improved mental health status in all patients, although the specific effectiveness of the program could not be identified. The majority of patients increased in performance during this rehabilitation program, although increases were small. For local clinics, this study presents a new perspective on exercise and its potential in psychiatric rehabilitation. It is recommended to individualize and specify exercise programs, especially regarding intensity, which provides potential for larger improvements in exercise performance, as it is suggested that such improvements may be positively related to depression symptoms.

## Figures and Tables

**Figure 1 jfmk-09-00233-f001:**
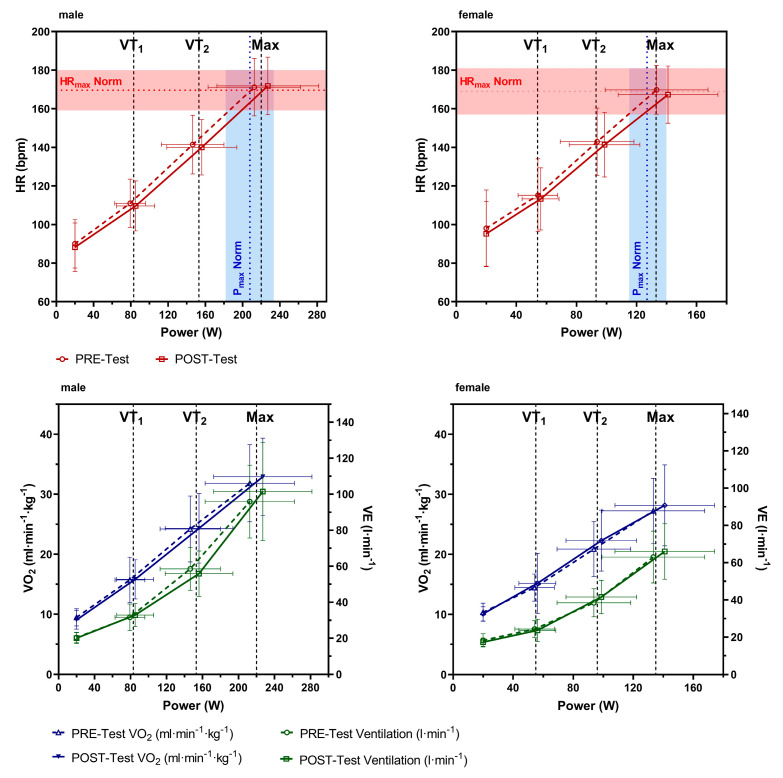
Heart rate (HR), oxygen uptake (VO_2_), and ventilation (VE) at the end of the 20 W warm-up and VT_1_ and VT_2_ and at maximum intensity of the cardiopulmonary exercise test performed pre- and post-rehabilitation for male and female participants. The red (HR_max_ norm) and blue dotted lines (P_max_ norm) represent the mean age-predicted HR_max_ and P_max_, respectively [38] The filled red and blue areas represent the standard deviation, respectively.

**Figure 2 jfmk-09-00233-f002:**
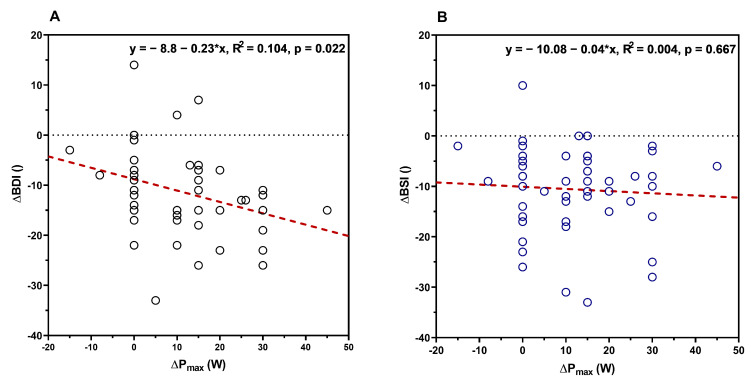
Relationship between pre-to-post-rehabilitation changes in maximal power (P_max_) from the incremental ergometer test and the pre-to-post changes in BDI (**A**) and BSI (**B**) scores.

**Table 1 jfmk-09-00233-t001:** General therapy content and scope.

Session	Application	Frequency
**Session** **1**	Medical massage	According to necessity
**Session** **2**	Physiotherapy and training therapy, respiratory therapy, body perception	3 U/week
**Session** **3**	Ergotherapy and other therapies	6 U/week
**Session** **4**	Psychotherapy and clinical and health psychology	9 U/week
**Session** **5**	Nutritional counselling	1 U
**Session** **6**	Indication-specific training	2 U/week
**Session** **7**	Others	According to necessity

U = one unit of 50 min or two units of 25 min.

**Table 2 jfmk-09-00233-t002:** Demographic data.

	Status	N ( )	N (%)
**Relationship**	In a relationship	31	58
	No relationship	22	42
Employed	Yes	36	68
	No	17	32
Inpatient stay due to psychical complaints in the last 12 months	Yes	12	23
	No	41	77
Main medication group	No	9	17
	Selective serotonin reuptake inhibitors	38	72
	Herbal sedatives	3	5
	Antipsychotic	2	4
	Statins	1	2

N ( ) = absolute number of participants, N (%) = number of participants relative to sample.

**Table 3 jfmk-09-00233-t003:** Anthropometric, clinical, and cardiorespiratory fitness measures of male, female, as well as for all participants pre- and post-rehabilitation, displayed as mean (±SD).

	All Participants			Male Participants			Female Participants		
Anthropometrics	Pre-	Post-	d	Pre-	Post-	d	Pre-	Post-	d
Age (yrs.)	41.0 ± 11.3			40.5 ± 10.6			41.8 ± 12.7		
Weight (kg)	77.4 ± 15.5	77.5 ± 14.9		85.4 ± 11.8	85.2 ± 11.0		64.3 ± 11.5	64.8 ± 11.3	
BMI (kg/m^2^)	25.3 ± 3.9	25.3 ± 3.7		26.6 ± 3.1	26.6 ± 2.8		23.0 ± 4.1	23.2 ± 4.0	
Clinical measures									
BDI ()	21.6 ± 8.3	10.1 * ± 9.5	1.31	21.4 ± 7.1	9.9 * ± 8.8	1.61	22.0 ± 9.6	10.5 * ± 10.8	1.04
BSI ()	65.1 ± 6.8	54.5 ± 11.3 *	1.23	65.3 ± 5.9	55.3 * ± 10.4	1.23	64.7 ± 8.0	53.2 * ± 12.7	1.23
Performance measures									
P_max_ (W)	182.6 ± 58.7	194.6 * ± 63.3	0.98	212.5 ± 49.7	227.0 * ± 54.6	1.11	133.4 ± 34.3	141.0 * ± 33.2	0.85
HR_max_ (bpm)	170.6 ± 14.0	170.1 ± 14.9		171.2 ± 15.0	171.8 ± 15.0		169.8 ± 12.6	167.3 ± 14.8	
VO_2max_ (mL·min^−1^·kg^−1^)	29.74 ± 5.92	31.09 * ± 6.91	0.48	31.82 ± 6.44	32.91 * ± 6.46	0.53	27.24 ± 5.42	28.16 ± 6.75	
P_VT1_ (W)	69.9 ± 19.5	74.2 * ± 22.7	0.59	79.3 ± 16.6	85.1 * ± 20.5	0.76	54.3 ± 13.1	56.1 ± 12.2	
P_VT2_ (W)	126.5 ± 39.7	134.5 * ± 43.2	0.85	146.4 ± 33.5	156.1 * ± 37.6	0.97	93.7 ± 24.6	98.7 * ± 23.6	0.66
HR_VT1_ (bpm)	112.6 ± 15.2	111.0 ± 14.2		111.0 ± 12.5	109.6 ± 13.0		115.1 ± 18.9	113.3 ± 16.1	
HR_VT2_ (bpm)	142.0 ± 16.0	140.5 ± 15.1		141.4 ± 15.2	140.0 ± 14.3		142.9 ± 17.4	141.4 ± 16.7	
VO_2VT1_ (L min^−1^)	14.50 ± 3.9	15.57 ± 3.98		15.81 ± 3.26	15.81 ± 3.26		14.46 ± 2.26	15.18 ± 5.01	
VO_2VT2_ (L min^−1^)	22.95 ± 5.37	23.55 ± 5.57		24.20 ± 5.50	24.31 ± 5.79		20.88 ± 4.57	20.05 * ± 5.18	0.56

* Significantly different (*p* < 0.05), d = effect size as Cohan’s d, yrs. = years of age.

## Data Availability

The data presented in this study are available on request from the corresponding author.

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
