# Peer review of "Changes in Exercise Performance in Patients During a 6-Week Inpatient Psychiatric Rehabilitation Program and Associated Effects on Depressive Symptoms"

_jfmk, 2024, doi:10.3390/jfmk9040233_

Round 1
Reviewer 1 Report (Previous Reviewer 2)
Comments and Suggestions for Authors
The authors have partially addressed concerns I raise in a previous review session.
Author Response
We thank the reviewer for the first round comments and changed the text where appropriate in this first round – regarding additional requests we are open for changes with a more detailed request. Regarding the methods section we revised the section with respect to other reviewers comments.
Reviewer 2 Report (Previous Reviewer 3)
Comments and Suggestions for Authors
he paper focuses on a training program designed to decrease the level and symptoms of depression.
This topic is vital, and the paper can be helpful to professionals.
The abstract's content is correct.
The Introduction provides a stable background for the research. The authors use global and European statistics. Since the program was carried out in Austria, the Austrian situation concerning people’s mental health may be added to these data. The purpose and aims are well-introduced. The relevance and necessity of the current paper can also be emphasised here. It would be good to have a hypothesis focusing on the efficacy of the training as well.
The materials and methods section should be restructured. The first paragraph (lines 118-139) can be part of a 2.1 Procedure subsection. Participants (lines 140-170) can be subchapter 3.2. After that, 3.3 Therapy program, 3.4 Cardiopulmonary exercise test (CPX), 3.5 Clinical measures and 3.6 Statistics. The content of these subchapters is professional and appropriate. Concerning the sample, some sociodemographic characteristics (e.g., gender, age, level of education).
The Results section introduces the findings appropriately. The content is clear and professional, and tables and figures support the understanding. The relationship between BSI and BSI can also be calculated and measured.
The content and order of the findings in the discussions are appropriate. The conclusions drawn are proper. The discussion of the findings should better reflect the hypotheses.
I believe that calculations and statistical data (e.g., lines 345-347, 352-354, 356-357, 404, 429, etc.) should be placed to the Results while the discussion is rather the interpretation of the findings.
The limitations mentioned are proper.
The Conclusions part is correct, too. Here, the practical implications should be introduced, along with the added value and novelty of the research.
Overall, this paper is a valuable study worth publishing after minor modifications.
Author Response
Dear reviewer,
Thank you for revising our manuscript and your valuable comments! Please find the respond to your comments below.
Comment 1
Since the program was carried out in Austria, the Austrian situation concerning people’s mental health may be added to these data.
Reply 1
We prescribed the European situation in the introduction and as the Austria situation is similar we kindly request not to include the local details (which are covered by the general European statistics).
Comment 2
The purpose and aims are well-introduced. The relevance and necessity of the current paper can also be emphasised here. It would be good to have a hypothesis focusing on the efficacy of the training as well.
Reply 2
Within the rehabilitation program patients undergo no detailed and controlled training program. Patients participated in group exercise sessions without a specific training regime. Therefore, we could only examine the effect of the program as a whole. However, since exercise sessions are unspecific and not on an individual level, we hypothesized that there will be no effects on performance of the standard rehabilitation program. However, we are fully aware of the fact, that individualized and specific training programs are necessary in the future. According to your suggestions we included this in the manuscript.
Comment 3
The materials and methods section should be restructured. The first paragraph (lines 118-139) can be part of a 2.1 Procedure subsection. Participants (lines 140-170) can be subchapter 3.2. After that, 3.3 Therapy program, 3.4 Cardiopulmonary exercise test (CPX), 3.5 Clinical measures and 3.6 Statistics. The content of these subchapters is professional and appropriate. Concerning the sample, some sociodemographic characteristics (e.g., gender, age, level of education).
Reply 3
Thank you for this comment. We revised the methods section according to your suggestion. Regarding the sociodemographic characteristics, we presented the available demographic data of the whole study sample in table 1 and 3 (age and sex). Unfortunately, we do not have access to information of participants highest education level.
Comment 4
The Results section introduces the findings appropriately. The content is clear and professional, and tables and figures support the understanding. The relationship between BSI and BSI can also be calculated and measured.
Reply 4
In the results section we present the changes of BSI values from pre to post, as well as the relationship between BDI and BSI pre and post rehabilitation: “Pearson correlations showed significant relationships between BDI and BSI at pre (r = 0.709, p < 0.001) and post time points (r = 0.740, p < 0.001)”.
Comment 5
The content and order of the findings in the discussions are appropriate. The conclusions drawn are proper. The discussion of the findings should better reflect the hypotheses.
I believe that calculations and statistical data (e.g., lines 345-347, 352-354, 356-357, 404, 429, etc.) should be placed to the Results while the discussion is rather the interpretation of the findings.
Reply 5
We agree that calculations are not supposed to be presented in the discussion and deleted some numbers which are not essential for the argumentation of the single points. (line 345-347, line 352-354, 404 rephrased, 429 -444 rephrased and numbers deleted, 467).
Comment 6
The Conclusions part is correct, too. Here, the practical implications should be introduced, along with the added value and novelty of the research.
Reply 6
Thank you, we expanded the conclusion according to your comment.
This manuscript is a resubmission of an earlier submission. The following is a list of the peer review reports and author responses from that submission.
Round 1
Reviewer 1 Report
Comments and Suggestions for Authors
This is an interesting, methodologically sound and well-written study with important topic and clinical implications regarding depression treatment. Although it was a one-arm, non-randomized study, thus susceptible to placebo effects (lack of control standard care group of depressive patients that did not undergo the exercise program aimed at improving CRF), it used a pre-post design, in which each participant served as a control of herself/himself.
I have some questions for clarification and some recommendations for improvement of the study:
Step-count per day reflecting physical activity mentioned in the abstract should be clarified whether referring to the beginning or the end of the rehabilitation program.
The conclusion stated also in the abstract that “The program significantly improved the mental health status of all patients” is arbitrary, since this could well be the effect of medications administered concomitantly, also considering that the time interval to the post-evaluation corresponds to the time newly –started antidepressants would kick in. Furthermore, psychotherapy could have played a role as well. This statement should be softened to correspond to the limitations of the study , e.g. “the program was associated with improved mental health status”.
At the end of paragraph 1, I would rather prefer the authors commented on exercise not as a replacement (at least yet) but as counterbalance to metabolic effects of medications
The following most recent meta-analysis on the subject should be added and its’ results discussed in the context of the present study Noetel et al 2024 Effect of exercise for depression: systematic review and network meta-analysis of randomised controlled trials (https://doi.org/10.1136/bmj-2023-075847)
In the first sentence of third paragraph ‘’ The level of cardio respiratory fitness (CRF) is an important measure of physical and mental status and a strong predictor of all-cause mortality [24] ‘’ how is the declaration of CRF as a measure of mental status supported by reference 24?
I don’t find the argumentation in the 3nd paragraph about the link between CRF and depressive symptoms convincing. CRF is not independent of training, so it is possible that both exercise performance and depressive symptoms improve through exercise in a separate way. Low fitness level and sedentary lifestyle is a characteristic of most depressed patients, so I think low CRF is more of a consequence rather than a cause of depression. Authors need to elaborate more on this.
Exclusion criteria should be fully described. I assume that except from cardiopulmonary disease patients with regular use of medications that limit blood pressure and heart rate, also patients with significant orthopedic problems and also other psychiatric diagnoses apart from depression and anxiety disorders were not included
Provide the formula how VΟ2 max was calculated
Provide justification for the selection of the tests used to measure depressive and other psychiatric symptoms
How were missing data from drop-outs dealt with?
Authors used paired-samples t-tests to show pretty much expected improvements in both CRF and depression post- vs pre-exercise. Regarding the effects of exercise on exercise performance, is there any difference between depressed patients and non-depressed individuals? Since there was no control-group in this study, there should be an attempt to compare effects sizes with published literature using similar exercise programs and age groups. The authors do state in the discussion that “increases in VO2max were lower than the suggested clinically meaningful changes of about 3.5 ml.min-1.kg-1 [53]”. Is there any more data on this, or also Pmax and VT1 and VT2?
A more illustrative comparison of the magnitude of the effects on depression with the results of reference studies 29, 30, or also 31 and 32 in depressed patients and also of factors that could account for possible variance would better inform the reader. Comparisons are indeed made in the discussion of the study with the study of Chu et al [54] and with ref 30 (of VO2peak), as well as comparisons with 29 and 31. In particularly compared with reference 29 (meta-analysis) however, more discussion is needed.
It would also be interesting to explore not only the change in measures of CRF as a predictor related to the change in depression score as the outcome measure but also run an additional regression model to look at the relationship of baseline measure of CRF as a predictor of following improvement in depression score.
A few more limitations of the study should be discussed such as: it used self-assessment tools to evaluate depression as opposed to physician –rated (patients more likely to report improvements simply due to their belief in the efficacy of the treatment), and it did not look at or had insufficient power to explore possible interactions between medications and exercise in improving depression.
Discussion on possible biological mechanisms by which amelioration of depressive symptoms could be associated with better physical fitness is missing
Although authors do state it should be individually tailored and more research is required, more comments on the type of exercise (aerobic, resistance training etc) and the intensity of the exercise (mild, moderate, vigorous) indicated for improving CRF and depression based on this study, but also on the literature is recommended
Comments on the Quality of English Language
There are only few language mistakes
Reviewer 2 Report
Comments and Suggestions for Authors
1. It is difficult to agree with the authors' claim (Almost no data are available on exercise performance), and we would like to see specific evidence for this.
2. Given that each subject has different initial exercise performance, it seems that the training program should be customized. Therefore, information on intensity is important, but the text states that the overall intensity was low to moderate. Please add more information on how this was done.
3. In the Discussion section, please add a discussion on why the effectiveness of the training was different between the BDI and BSI.
Reviewer 3 Report
Comments and Suggestions for Authors
The paper focuses on a 6-weeks standard stationary rehabilitation on performance and the degree of depression focusing on psychiatric patients, introducing the results of an empirical research. This topic is very important and the paper can be useful for professionals.
The abstract's content is corrrect.
The information provided in the Introduction is appropriate. The paragraph referring to the purpose and aims can be improved. The relevance and necessity of the current paper can also be emphasised here along with the research questions and /or hypotheses.
The materials and methods section is detailed. The authors should introduce some core socio-demographic data, e.g. Gender, age, highest education, parental characteristics (if available). Procedure, instruments and statistical analysis are professional and appropriate.
The Results section introduces the findings appropriately. The content is clear and professional, and tables and figures support the understanding. The 3.2 Activity subsection is too short, it should be integrated to 3.1 When added to the Introduction charter, the results should be presented in the order of the hypotheses.
In the Discussions section, the arrangement and substance of the findings are suitably presented. The authors arrive at valid conclusions based on the results obtained. Upon formulating the hypotheses, it is essential for them to engage in reflection regarding these hypotheses. It is imperative to juxtapose the present findings with earlier research documented within this paper, emphasizing the unique contributions and innovations that this current study offers in comparison to previous works. Nonetheless, this comparison must be executed by deriving relevant conclusions that refer to the research questions and/or hypotheses.
In the end, the limitations mentioned are correct.
The Conclusions part is correct too. Here, the practical implications should be introduced.
Overall, this paper is a valuable study worth publishing after minor modifications.